# Semiconductor Nanomaterial Photocatalysts for Water-Splitting Hydrogen Production: The Holy Grail of Converting Solar Energy to Fuel

**DOI:** 10.3390/nano13030546

**Published:** 2023-01-29

**Authors:** Muhammad Mohsin, Tehmeena Ishaq, Ijaz Ahmad Bhatti, Asim Jilani, Ammar A. Melaibari, Nidal H. Abu-Hamdeh

**Affiliations:** 1Department of Chemistry, University of Agriculture Faisalabad, Faisalabad 38040, Pakistan; 2Department of Chemistry, University of Lahore, Sargodha Campus, Sargodha 40100, Pakistan; 3Center of Nanotechnology, King Abdulaziz University, Jeddah 21589, Saudi Arabia; 4Department of Mechanical Engineering, King Abdulaziz University, Jeddah 21589, Saudi Arabia; 5Center of Research Excellence in Renewable Energy and Power System, King Abdulaziz University, Jeddah 21589, Saudi Arabia

**Keywords:** nanomaterial, hydrogen production, water splitting, semiconductor materials, photocatalysis, green source

## Abstract

Nanomaterials have attracted attention for application in photocatalytic hydrogen production because of their beneficial properties such as high specific surface area, attractive morphology, and high light absorption. Furthermore, hydrogen is a clean and green source of energy that may help to resolve the existing energy crisis and increasing environmental pollution caused by the consumption of fossil fuels. Among various hydrogen production methods, photocatalytic water splitting is most significant because it utilizes solar light, a freely available energy source throughout the world, activated via semiconductor nanomaterial catalysts. Various types of photocatalysts are developed for this purpose, including carbon-based and transition-metal-based photocatalysts, and each has its advantages and disadvantages. The present review highlights the basic principle of water splitting and various techniques such as the thermochemical process, electrocatalytic process, and direct solar water splitting to enhance hydrogen production. Moreover, modification strategies such as band gap engineering, semiconductor alloys, and multiphoton photocatalysts have been reviewed. Furthermore, the Z- and S-schemes of heterojunction photocatalysts for water splitting were also reviewed. Ultimately, the strategies for developing efficient, practical, highly efficient, and novel visible-light-harvesting photocatalysts will be discussed, in addition to the challenges that are involved. This review can provide researchers with a reference for the current state of affairs, and may motivate them to develop new materials for hydrogen generation.

## 1. Introduction

Environmental pollution is one of the most dangerous and destructive factors nowadays [1,2,3,4]. The major cause of environmental pollution is the rapid increase in greenhouse gas emissions, such as CO_2_, CH_4_, N_2_O, and chlorofluorocarbons, into the atmosphere [5,6,7]. Among the various sources, the most influential source of greenhouse gases is the consumption of fossil fuels for energy production [8,9]. The content of greenhouse gases is increasing because of the progressive consumption of fossil fuels over recent decades. With an increase in population rate, there is an increase in fuel consumption, resulting in an enhanced rate of greenhouse gas emissions, which adds more pollution to the environment. Moreover, fossil fuels are not a satisfactory source of the present energy demand because of their continuous depletion. Thus, an alternative energy fuel must be developed to fulfill the future energy requirements [10,11]. In this regard, hydrogen emerges as a potential candidate to meet the world’s energy demand because it is the best energy carrier [12,13]. Hydrogen retains a higher energy-harvesting aptitude per unit mass, 122 kJ/g, than gasoline, 40 kJ/g [14]. Hydrogen is the most abundant naturally occurring element in the universe, both in the free and combined states [15]. Hydrogen production via the photochemical splitting of water has attracted the attention of researchers after the pioneering work of Honda and Fujishima on photoelectrochemical cells in 1972 [16]. Water is an effortlessly available and abundant source of hydrogen [17], and by using a suitable semiconductor photocatalyst, it can be split into hydrogen and oxygen in the presence of sunlight [3,18]. Hydrogen-based energy systems can be regarded as a long-term solution to environmental problems and a sustainable solution to future energy demands [19]. 

Hydrogen can be produced from many sources. The primary hydrogen sources consist of hydrocarbons, including naphtha, methanol, heavy oil, natural gas, coal, and biomass. Hydrogen can be produced via the gasification and reforming of hydrocarbons, expressed by Equations (1) and (2).
CH_4_ + H_2_O → CO + 3H_2_O(1)
CO + H_2_O → CO_2_ + H_2_(2)

The production of CO_2_ during steam reforming limits the efficiency and applicability of the process since it causes global warming and increases environmental pollution, which is of serious concern. Since water is abundantly available on earth, it can be a convenient resource for cost-effective hydrogen production. However, it requires energy to split water. Depending on the power source, water splitting can be categorized into three groups: (i) thermochemical water splitting, (ii) photobiological water splitting, and (iii) photocatalytic water splitting [20].

Among these three approaches, photochemical water splitting is more useful because it can produce hydrogen without any by-product, it utilizes inexpensive instruments, and it has a higher efficiency with large-scale hydrogen production. Figure 1 shows the basic principle of the overall water-splitting mechanism, which consists of a few successive steps: (i) light absorption, (ii) excitation of electrons from the ground state to an excited state, (iii) transfer of charges to the surface of the photocatalyst, and (iv) surface reactions between photogenerated charges and water molecules.

Solar energy is used to split water in the presence of a suitable semiconductor photocatalyst (Equation (3)).
(3)$$2H_{2}O\;\overset{Photocatalyst+sunlight}{\to }\;2H_{2}+O_{2}$$

Honda and Fujishima are the pioneers of this research, and they used TiO_2_ as a photocatalyst to split the water in the presence of sunlight [4]. The photocatalyst used in water splitting is responsible for the conversion efficiency of water to hydrogen. Considerable work has been carried out to develop an efficient semiconductor photocatalyst, maximizing the efficiency for enhanced hydrogen production. Although numerous reviews regarding photocatalytic hydrogen production have been published in the literature [22,23,24], only a few of them describe the comprehensive and collective description of hydrogen production pathways [20], modification tactics for the enhanced production rate of H_2_ [25,26], and photocatalytic activity of various transition-metal-based photocatalytic families. The current review highlights the various production pathways for hydrogen generation by applying various energy forms in terms of thermal, solar, and electrical power in thermochemical, photochemical, and photoelectrochemical water splitting, respectively. Furthermore, the various modification strategies used to enhance the catalytic efficiencies and kinetic rates of semiconductor photocatalysts are described. Additionally, this review considers the progress of various transition-metal-based photocatalysts, including titanium, tantalates, niobates, metal nitrides, metal oxynitrides, and metal sulfides. Moreover, the key challenges and future perspectives are summarized, which can offer insights into useful directions for future developments to fulfill the growing energy needs.

## 2. Hydrogen Production Pathways

Hydrogen production has become an important research topic worldwide [27]. In this regard, several methods have been employed to check the feasibility of hydrogen production. Many processes, such as thermochemical processes, electrochemical processes [28], direct solar water splitting, and biological processes, have been introduced with different efficiencies and limitations.

### 2.1. Thermochemical Processes

Thermochemical processes use heat for chemical reactions to transform fossil fuels and biomass into hydrogen. Some important techniques in thermochemical processes are (i) steam methane reforming (SMR), (ii) coal gasification, and (iii) biomass gasification. In the SMR technique, methane gas is treated with steam in the presence of a suitable catalyst at high temperatures (100–1000 °C) [29]. This process is endothermic because energy is required to start and complete the reaction (Equation (4)).
CH_4_ + H_2_O (steam) → CO + 3H_2_(4)

Subsequently, carbon monoxide reacts with water to produce more hydrogen, which is known as the water–gas shift reaction (Equation (5)).
CO + H_2_O → CO_2_ + H_2_(5)

Only a small amount of carbon dioxide is produced but it can be captured and stored using various separating techniques [30]. The main advantage of the SMR approach is that it produces one-half of the greenhouse gases compared with that of other fossil fuels, thus minimizing the pollution rate and maintaining ecological sustainability [14]. Moreover, coal can be used to produce hydrogen. Coal is treated with oxygen and steam under high temperatures and high pressure to produce hydrogen and carbon monoxide. Carbon monoxide further undergoes a water–gas shift reaction to produce more hydrogen. Hydrogen is captured using a separating system, and CO_2_ is also captured and stored [31]. Here, hydrogen is produced using the same principle involved in SMR processes. Biomass reacts with steam at high temperatures and pressure to produce hydrogen and carbon monoxide and this carbon monoxide again reacts with water to yield more hydrogen.

Thermochemical water splitting requires a high temperature to split the water chemically into its constituents, and the splitting efficiency is also low compared with photocatalytic water splitting. This method has the advantage of low or no greenhouse gas emissions. However, the high temperature is difficult to handle and the large-scale production of hydrogen cannot be achieved, owing to the instrument’s high cost.

### 2.2. Microbial Biomass Conversion

In this process, bacteria and algae are used to convert and digest the biomass and release hydrogen on a small scale, with no environmental impact [32]. Research on this process is also in the early stages, so the development for hydrogen production requires more work [31].

### 2.3. Electrolytic Processes

This process uses electricity to produce hydrogen and oxygen from water-based electrolytes and works according to the principle of electrolysis. The oxidation of $O^{2-}$ ions takes place at the anode to release oxygen, and the reduction of 2$H^{+}$ takes place at the cathode to release hydrogen. In this process, hydrogen is produced with zero greenhouse gas emission, but electricity is required to drive the whole process, and the cost and production method limit the electrolytic process for hydrogen production [33,34].

### 2.4. Direct Solar Water Splitting

This strategy involves two major processes. Generally, bacteria and microorganisms are used to split the water into its constituents in the presence of sunlight. Hydrogen production through biological processes requires more research because the development is still in an earlier stage. Currently, this process is limited because of low efficiency [14]. Photoelectrochemical water splitting is the most feasible [35] and affordable process to produce hydrogen from renewable resources [36]. This process uses solar energy in the form of radiation to split water into hydrogen and oxygen on the surface of a suitable semiconductor photocatalyst [12,37]. Here, the photocatalyst is immersed in water, and this water undergoes a splitting mechanism when the semiconductor surface is subjected to solar irradiation of a suitable wavelength, resulting in the production of hydrogen [38]. Ultimately, this process offers a high solar-to-hydrogen conversion efficiency and a long-term, sustainable solution to future energy demand with almost zero greenhouse gas emissions. This approach is more significant among all the above-described approaches for hydrogen generation, but still requires more research to enhance the process efficiency [39,40].

## 3. Basic Principle of Water Splitting

From a thermodynamic perspective, photocatalytic water splitting is an endothermic process that involves a large positive change in Gibbs free energy, approximately 237 kJ/mole. Photocatalytic water splitting uses semiconductor photocatalysts because semiconductor substances are capable of absorbing light or photons that excite their electrons from the valence band (VB) to the conduction band (CB). As a result, electron–hole (e/h) pairs are generated, which can react with water and split it into their constituents to form hydrogen and oxygen.

Semiconductors absorb photons of energy equal to or greater than the energy band gap (Eg) that exists between the VB and CB. When it absorbs a photon, electrons from VB are promoted to CB by creating holes (h^+^) of positive charge in the VB, and the promotion of an electron to CB results in a negative charge. This e/h pair is used to split water into hydrogen and oxygen via oxidation and reduction. Semiconductors should be of a smaller band gap to absorb visible light for higher efficiency. If the band gap is large, then UV light is absorbed by a photocatalyst, which limits its efficiency by 3–5% because solar radiation has a UV content of less than 5%.

Two types of configurations are used to produce H_2_ and O_2_ from photocatalytic water splitting: (i) photoelectrochemical cells and (ii) particulate photocatalytic systems. A photoelectrochemical cell consists of two electrodes: one of them is a semiconductor photocatalyst dipped in an aqueous or water-based electrolyte [41]. As the system is irradiated, the photocatalyst produces e/h pairs, where the h^+^ is used to oxidize water and the e^-^ is used to reduce the 2H^+^ ions into H_2_ gas. Figure 2 shows the progression of these e/h pairs during this electrolytic process.

The particulate photocatalytic system involves tiny electrodes in the form of small particles or granules of semiconductor material immersed in a water-based electrolyte. When light strikes the system, the subsequent processes occur: (i) absorption of light and generation of charge carriers, (ii) charge carriers move from the bulk of the photocatalyst to the surface of the catalyst, and (iii) chemical reaction between water and photogenerated charge carriers. In a particulate photocatalytic system, water is adsorbed on the surface of a photocatalyst. After the second step, when charge carriers are generated on the surface, a redox chemical reaction takes place, such that water is split into oxygen and hydrogen (Equations (6) and (7)), as shown in Figure 3.
(6)$$H_{2}O+h^{+}\;\to \;2H^{+}+\frac{1}{2}O_{2}$$
2H^+^ + e^−^ → H_2_(7)

The overall reaction is:(8)$$H_{2}O\;\to \;H_{2}+\frac{1}{2}O_{2}$$

**Figure 3 nanomaterials-13-00546-f003:**
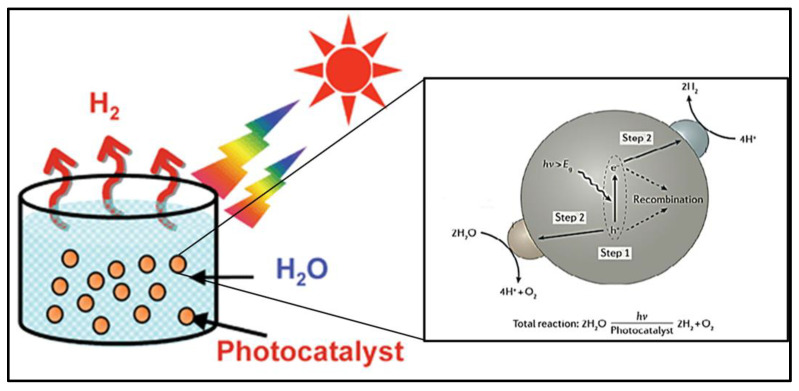
Shows a schematic diagram of hydrogen production from a particulate photocatalytic system. Printed with permission from the Royal Chemical Society [43].

Particulate photocatalytic systems have a few limitations, e.g., charge carrier separation is difficult as compared with separation in photoelectrochemical cells and difficulties exist in separating the produced gases to prevent the reverse reaction. However, it is simple and easy to produce hydrogen using a particulate photocatalytic system. Some reducing agents, such as alcohols, sulfides, sulfites, and ethylenediaminetetraacetic acid, and oxidizing agents, such as persulphates, Ag^+^, and Fe^+3^, are also used to facilitate photocatalytic water splitting [40,43].

## 4. Merits for Efficient Photocatalysts and Associated Challenges

To achieve maximum solar light harvesting in the visible part of the solar spectrum, and higher yield in the water-splitting process, it is necessary to develop a photocatalyst with the desired band gap energy and electrochemical properties. It must be capable of absorbing visible light to achieve maximum efficiency. In this regard, several binary metal oxides, metal-nonmetal composites, and doped metal alloys are used to check their ability to convert visible light to chemical energy based on their light absorption capacity, the kinetics of charge transfer, and electronic structure. 

The material used as a photocatalyst in a particulate photocatalytic setup must possess several useful properties. For example, it should absorb sufficient solar radiation, especially in the visible region, to maximize efficiency. Furthermore, the band edge potential of a semiconductor catalyst should be appropriate for water splitting. It should not allow the photogenerated charge carriers to recombine, resulting in energy losses, and must be stable toward chemical corrosion and photo-corrosion. Kinetically, it should facilitate the transfer of photogenerated electrons from the bulk of the photocatalyst to the surface, to oxidize the water. To carry out water oxidation, the bottom of the CB must be at a more negative potential than that of the reduction potential of protons, known as 2H^+^/H_2_, on a normal hydrogen electrode [44] scale at pH = 0. To oxidize the water that is already adsorbed on the photocatalyst surface, the oxidation potential of the valance band edge must be higher than the oxidation potential of water (v = +1.23 eV). Therefore, 1.23 eV is the minimum energy required to achieve water splitting, corresponding to a wavelength of 1010 nm. Hence, approximately 70% of all solar photons are available to carry out energy-based processes such as water splitting. Table 1 shows the solar spectrum’s energy distribution, their corresponding wavelength, and their contribution to the total spectrum.

However, there are several unavoidable energy losses, such as e/h recombination. The band gap of a semiconductor is determined by its electronic structure, which is determined by its constituent atoms and bonding. Band positions for several semiconductor photocatalysts are shown in Table 1 [45].

Among many semiconductor photocatalysts, the band position of KTaO_3_, SrTiO_3_, TiO_2_, ZnS, CdS, and SiC fulfill the thermodynamic requirements for the overall water-splitting reaction. Another problem associated with a variety of photocatalysts, such as CdS and GaP, is that they can become oxidized when they absorb light, rather than oxidizing the water, which limits their efficiency for water splitting. The selected photocatalyst for water splitting must be capable of preventing the chemical reaction at the photocatalyst water interface to prevent electrochemical corrosion, photo-corrosion, and dissolution [40,46,47]. 

Furthermore, the mismatch among the band position of narrow-band-gap catalysts and the redox aptitude of water is also a key challenge, which limits the catalytic performance of semiconductors on a commercial scale. The valence band maximum (VBM) and conduction band maximum (CBM) of the semiconductor photocatalyst must straddle water’s oxygen evolution and hydrogen evolution aptitudes to achieve enhanced water-splitting performance [7]. 

Moreover, narrow-band-gap semiconductors comprising N^3−^ and S^2−^ are prone to oxidation in the presence of oxygen during the oxidation step in the redox reaction of water. Owing to these challenges, only a few semiconductors are used in overall photocatalytic water splitting in the visible region of the solar spectrum. Thus, new approaches are being applied to achieve enhanced photocatalytic performance by focusing on the half-reactions, either oxygen evolution or hydrogen evolution. In this approach, sacrificial agents, either electron acceptors or donors, are correspondingly used for oxygen evolution or hydrogen evolution [8].

Two-step excitation, or the Z-scheme, is also applied to tackle this challenge, in which oxygen and hydrogen are liberated by two separate semiconductor catalysts, which are the oxygen evolution catalyst (OEC) and hydrogen evolution catalyst (HEC). The OEC and HEC fulfill the compulsory thermodynamics for their respective half-reactions in the splitting of water, and thus enable researchers to use a wider variety of semiconductor photocatalysts. Such systems have been widely designed and evaluated. One such setup was designed by Qi et al., in which barium-functionalized Ta_3_N_5_ was used for the HECs, and PtOx/WO_3_ was used as an OEC [9].

## 5. Effective Ways to Engineer Efficient Photocatalysts

Properties of photocatalysts, such as small band gap, e/h pair generation, and e/h pair segregation, are the prerequisites for higher hydrogen generation activity. Although many catalysts have been reported, none of them are ideal to fulfill the growing energy needs at the industrial scale because it is difficult to find a single material that possesses all the necessary properties. The ratio of hydrogen production determines the efficiency of a photocatalyst to absorbed solar energy in the form of radiation, and until now, the maximum efficiency (quantum yield) achieved by using the Rh_2-Y_Cr_Y_O_3_/(Ga_1-X_Zn_X_) (N_1-X_Zn_X_) photocatalyst to produce hydrogen is limited to 5.9%. This catalyst is applied to clean water under visible-light irradiation with wavelengths of 420–440 nm using a 300 W xenon lamp with a cut-off filter [48]. The low efficiency of the process is still a major limitation that restricts its application on a large scale.

Time-resolved spectroscopic techniques reveal that charge carriers generated via photocatalyst irradiation will recombine; nearly 90% of charge carriers are recombined immediately after they are generated, while only 10% of charges appear at the surface to oxidize the water and reduce the hydrogen ions into hydrogen gas [49].

To develop an efficient photocatalyst, researchers must control the interdependence of the electronic, surface, and microstructural properties. In this regard, many efforts have been made to develop them, such as (i) tuning of UV-active photocatalysts (related to band gap engineering), (ii) surface modification of photocatalyst by deposition of co-catalyst to reduce the energy of activation for gas evolution, (iii) sensitization, and (iv) nano-designed photocatalysts for control over the morphology and defects. Researchers apply these processes to enhance the efficiency of photocatalysts by reducing the recombination of photogenerated charges.

To date, one-dimensional (1D) nanomaterials with open channels have received significant attention owing to their decreased mass transfer resistance, reduced diffusion distance, enhanced photogenerated carriers separation, and increased light absorption [50]. For example, Wenjun Wang et al. fabricated 0D/1D mixed-dimensional black phosphorus/carbon nitride nanohybrids using simple ultrasonic methods. The resultant hybrid revealed enhanced photocatalytic performance owing to efficient charge segregation facilitated by black phosphorus quantum dots [51].

### 5.1. Band Gap Engineering

Band gap engineering involves the conversion of a UV-active photocatalyst to a visible-light photocatalyst to capture solar radiation to a maximum extent. It can be performed in two ways: (i) cation doping or anion doping in a semiconductor lattice and (ii) the use of semiconductor alloys. Frequently, doping has been used to develop visible-light activity in UV-active substances (photocatalysts) [52]. Because of the large band gap in photocatalysts, they absorb high energy radiation (UV radiation), corresponding to approximately 4% of the total solar spectrum. Transition metals, such as antimony, tantalum, chromium, and zinc, as well as carbon, can also be used as dopants in semiconductors to enhance the photocatalyst response to visible light [53].

### 5.2. Doping

A change in electronic structure can be achieved by doping or replacing a cation with another element. Doping results in the contraction of the band gap in semiconductors by injecting energy levels as impurities between the VB and CB of semiconductors. This facilitates the photogenerated charge carriers to move to the surface to oxidize and reduce the water, which enhances its efficiency by bringing it to the visible portion of the solar spectrum. Although the reactivity to visible light increases, cation doping provides active centers or sites for photogenerated charge carriers to recombine instead of oxidizing the water [54]. The energy levels injected by cation doping within the semiconductor band gap hinder the movement of holes and electrons from going to the surface of the catalyst to perform the redox reaction [55]. Cation-doped semiconductor photocatalysts should be fine-tuned to split the water and maximize quantum yield. Many chemical techniques, such as precipitation and impregnation, are used to dope different metals into the host structure. Figure 4 shows the visible-light response of a cation-doped wide band gap structure [56].

Doping often leads to defects in the oxidation state of oxygen which can act as recombination centers for photogenerated charge carriers. Figure 5 evaluates the anion-doping effect on the semiconductor’s band gap. Anion doping is an alternative method to develop a visible-light response in large band gap UV-active photocatalytic materials. Numerous researchers are interested in anion doping in semiconductor structures because of its feasibility, and it does not provide any active centers for photogenerated charge carriers to recombine as they are formed. Different anions, such as N [58,59], can be doped in oxide semiconductors. An oxide semiconductor photocatalyst has a large band gap in which the 2p orbital of oxygen is on the top of the VB. Anion doping results in the mixing of the 2p atomic orbital of the anion with 2p of oxygen to decrease the band gap energy of the wide-bandgap semiconductor, as shown in Figure 6. This process raises the VB closer to the CB and effectively makes the UV-active semiconductor into a visible-light-active photocatalyst.

### 5.3. Semiconductor Alloys

In this technique, two semiconductors, one that has a wide band gap and that has a small band gap, are mixed to give a solid solution in which lattice sites of both semiconductor materials are interspersed to make the photocatalyst more responsive to visible light, as depicted in Figure 6. By varying the composition of solid solution, the band gap of mixed semiconductor photocatalysts can be adjusted. Examples of such alloys include GaN-ZnO [60] and ZnS-CdS [61].

**Figure 6 nanomaterials-13-00546-f006:**
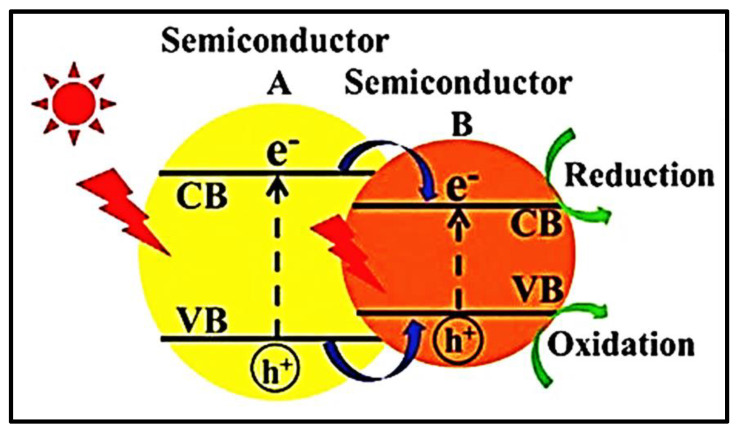
Representing the band gap of material obtained via a solid solution of wide and narrow-band-gap materials. Printed with permission from Wiley Online Library [62].

### 5.4. Surface Co-Catalyst

Photocatalytic water splitting involves surface chemical reactions between photogenerated charge carriers and water to produce H_2_, as discussed above. Hydrogen production via water splitting and surface adsorption of charges can be enhanced using noble metals or their oxides by depositing them on the surface of a photocatalyst as a co-catalyst, e.g., Rh, Pt, RuO_2_, and NiO_2_. Co-catalysts enhance the rate of surface reactions by capturing the photogenerated holes and electrons and preventing them from recombining. This makes it easier for holes to oxidize water and reduce the hydrogen ions by lowering the activation energy and, thus, enhancing the visible-light activity of the UV-active photocatalyst [60], as shown in Figure 7.

### 5.5. Nanostructure

Photogenerated charges determine the efficiency of a photocatalyst, where either they recombine or go to the surface of the photocatalyst. The properties of the photocatalyst are responsible for the hydrogen production and recombination or transfer of the charges. Assets, such as surface properties, crystal size, crystallinity, and structural defects, are responsible for transferring photogenerated charges to the surface of a photocatalyst or allowing them to recombine. At the nanoscale, the crystalline structure of the photocatalyst behaves differently compared with the bulk scale. At the nanoscale, the crystal size, crystalline arrangements, crystal imperfections, and other surface properties greatly affect charge transportation. Reduction in the size of the photocatalyst allows the photogenerated charges to move to the surface more easily, increasing efficiency [64]. Higher crystallinity leads to a higher charge transfer rate, although it is necessary to control structural imperfections because they stimulate charge reunion [65]. New electronic states are introduced at the nanoscale, in the band gap, which prevents charge recombination. Therefore, the photocatalyst should be designed at the nanoscale because it enables control of the band gap by altering the particle size [25]. 

Efforts have been made to study the particle size effect on light absorption, movement of photogenerated charges, and surface area, which leads to the synthesis of the materials at the nanoscale with controlled particle size, structure defects, and dimensions. Materials can be designed at the nanoscale using various methodologies. Nonconventional technologies, such as sol–gel [66], hydrothermal [67], micelles and inverse micelles [68], chemical vapor deposition (CVD) [69] and sonochemical technique [70], have been developed to synthesize nanoparticles with controlled dimensions and morphologies and improve photocatalytic efficiencies of photocatalysts. The problem of agglomeration is associated with these nanoparticles because of inter-particle interactions. Therefore, a microporous framework (zeolites [71,72] and activated carbon) is introduced during the process, which acts as a host for nanoparticles and prevents agglomeration. The three-dimensional structure of nanoparticles also contributes to this issue. Wang et al. [73] reported the superior activity of the colloidal TiO_2_ nanoparticles when they form a three-dimensional network along a given crystallographic plane. The authors proposed a so-called antenna effect, in which an antenna system or pathway is responsible for transferring charge carriers to the surface of a photocatalyst. Recently, powdered-type photocatalysts are used, which carry strong electronic coupling forces between particles and are enough to agglomerate, resulting in microscale particles. At the microscale, antenna effects arise, such that the rate of transfer of charge carriers increases, which increases the efficiency of the process. Apart from this, smart molecular engineering can also be applied to synthesize the photocatalyst at nanoscale, which has more photoactivity than the present system.

### 5.6. Multiphoton Water Splitting

In this process, two semiconductor photocatalysts are used to split the water into its major constituents [74,75]. Positive charge carriers from one photocatalyst are used to oxidize the water, and negative charge carriers from the second photocatalyst are used to reduce the hydrogen ions to hydrogen gas. Semiconductor photocatalysts applied to water should have a small band gap, and water gets simultaneously oxidized and reduced via a two-step water-splitting reaction [64,76], as depicted in Figure 8.

Oxygen evolves at one semiconductor photocatalyst while hydrogen is evolving from another. However, controlling the recombination reactions between the generated charges is challenging. Therefore, the process is limited until a suitable solution to this problem is proposed [77].

## 6. Recent Breakthroughs in the Field of Photocatalysts

### 6.1. Z-Scheme Heterojunctions

Recently, the Z-scheme has been applied to enhance the photocatalytic activity of semiconductors by mimicking the natural photosynthesis in plants, which comprise two photosystems, PS-1 and PS-2. This scheme improves the redox potential of photocatalysts and enhances charge segregation aptitude. In the Z-scheme photocatalytic setup, two semiconductors (which can be labeled as PC-1 and PC-2) with compatible intermediate pairs and staggering band configurations, such as I^3−^/I^−^, Fe^3+^/Fe^2+^, and IO^3−^/I^−^, are applied [78]. One of the materials is an oxygen-evolving catalyst, while the other is a hydrogen-evolving catalyst, to execute total water splitting (TWS). Photoprovoked electrons in hydrogen-evolving catalysts reduce the water to produce H_2_ on the catalytic surfaces while holes play a crucial role in oxidizing the reducing agent to generate an oxidant. Moreover, the oxidized agent is again reduced on the catalytic surface of the oxygen-evolving catalyst, and oxygen is liberated, owing to water oxidation accomplished by holes. This type of photocatalytic setup for water splitting reveals maximal activity for the higher hydrogen yield because charge reunification is significantly reduced [20].

The charge carrier mobility and operation pathway are generally evaluated according to the band edge value; for example, a simple and novel Z- scheme photocatalytic semiconductor hybrid was fabricated by immobilizing MIL-101(Fe) on the graphitic surface of carbon nitride (g-C_3_N_4_), which revealed higher charge segregation aptitude and improved photocatalytic activity. Furthermore, co-catalysts and morphological modulation can also improve charge segregation. For instance, Kong et al. designed a 3-D MoS_2_-decorated UiO-66-(COOH)_2_ ZnIn_2_S_4_ heterostructure, as shown in Figure 9. In this ternary nanohybrid catalyst, nanoparticles of UiO-66-(COOH)_2_ were dispersed irregularly, and nanosheets of MoS_2_ were added to the edges of ZnIn_2_S_4_ nanosheets, which facilitated the contact intimacy of multiple structures to enhance the kinetics of charge mobility. The optimum photocatalytic hydrogen production rate was found to be 18.790 mmol/h/g in the UV–visible region [79]. 

Although it is best to enhance the photocatalysts’ charge segregation capability and improve catalytic activity, a few factors limit the efficacy of the Z-scheme-based photocatalytic systems. First, it is restricted to solution-phase reactions only. Second, side reactions can also occur, slowing the kinetic of water splitting and lowering the hydrogen yield. 

### 6.2. S-Scheme Heterojunctions

Since the Z-scheme faces some critical shortcomings, it is necessary to search for an alternate setup and introduce a new approach where the catalytic mechanisms can be clearly evaluated. J. Fu et al. introduced a new approach in which they described a new setup called step-scheme (S-scheme) heterojunctions [80].

The S-scheme heterojunction catalyst is chiefly composed of two n-type photocatalysts labeled PC-1 and PC-2, which perform oxidation and reduction, respectively. The transference of photoprovoked electrons is similar to n-type or step-like transfer, and all the holes and electrons are correspondingly segregated in the VB of PC-1 and the CB of PC-2. The internal electric field between PC-2 and PC-1 is the main driving power for the charge carrier relocation and mobility. Generally, the PC-1 (oxidation photocatalyst) has a lower Fermi level and higher work function, whereas the opposite is true for PC-2 (reduction photocatalyst). When PC-2 and PC-1 interact, the electrons from PC-2 are transferred to PC-1 through their interface, giving rise to electric field generation in the direction from PC-2 to PC-1. In S-scheme setups, somewhat unusable electrons in the CB and quite useless holes in the VB of PC-1 and PC-2, respectively, are reunited and eradicated at the interface. Conversely, the useful holes in the VB and expedient holes in the CB of PC-1 and PC-2, respectively, are retained owing to the existence of an electric field. Lastly, photocatalytic oxidation and reduction processes are simultaneously conducted by electrons in the CB and holes in the VB of PC-2 and PC-1.

Fortuitously, such heterojunctions reveal considerable practicality in spatial charge segregation. For instance, a typical 2D-2D setup comprising WO_3_ and g-C_3_N_4_ was fabricated using an electrostatic self-assembly method, which showed superior photocatalytic performance for hydrogen production (Figure 10) [81].

### 6.3. Metal–Organic Frameworks (MOF)

Metal–organic frameworks (MOF) are a novel and new class of extremely porous materials consisting of metal ions and organic ligands with long-range order and super crystallinity. Because of many inherent characteristics, including distinct topologies, higher surface areas, adjustable pore structure, and flexible chemical components, a large variety of MOFs are applied for catalytic water splitting. Furthermore, their catalytic assets can be augmented by adding different functional moieties, e.g., carbon nanotubes (CNT), metal compounds, polyoxometalates (POM), and other conductive substances, to make MOF/substrates or guests/MOFs. The high electrocatalytic activity for water splitting is credited to superconductivity and a large number of active sites [82]. Moreover, these assets can be improved by functionalizing or tuning their structures in terms of functionalizing nodes, linkers, and pores and incorporating more catalytic active sites while conserving their higher porosity [83].

Mori et al. used MOFs as catalysts for photocatalytic water splitting in 2009 [84], and since then, the work on this research area has thrived, which includes several review articles. For instance, Xu et al. reviewed MOF-based photocatalysts for water splitting to generate hydrogen in 2018 [85]. Shi et al. designed a high-nuclear Cu_24_-based POM@ MOF nanohybrid, the first noble-metal-free catalyst for TWS, playing a dual role to produce oxygen and hydrogen with superior photocatalytic activity [86]. Notably, [Cu^I^_24_(µ_3_-Cl)_8_ µ_4_-Cl)_6_] offered the perfect site for Cu^I^_24_ /MOF hybrid surface interaction, increasing the kinetics of hydrogen evolution and the regeneration of ZZULI-1. Furthermore, the synergic photocatalysis between two POM types, in addition to larger spaces in the ZZULI-1 channels, enhanced the oxygen liberation. The usage of hybrid ZZULI-1/PMOs aims to attain a highly proficient charge transfer mechanism via dual-functionalized ZZULI-1 and oxidized [Ru (bpy)_3_]^3+^, restoring the oxidized [Ru (bpy)_3_]^3+^ to its initial form (Figure 11) [86]. 

Kampouri et al. aimed to enrich the photocatalytic performance of MOF by fabricating MOF-MOF heterojunctions consisting of MIL-167 and MIL-125-NH_2_ [87]. The MIL-167/MIL-125-NH_2_ composites revealed the characteristic of type-II heterojunctions, or staggered gaps (Figure 12). Consequently, MIL-125-NH_2_ and MIL-167 played the roles of photocatalysts and photosensitizers, respectively. MIL-167 absorbs in the extended visible spectrum, i.e., above 500 nm of solar light, causing photoexcitation of electrons from the VB to the CB, and these electrons are transferred to the CB of the MIL-125-NH_2_. The electron transported from MIL-167, in addition to those exposed to photoexcitation in MIL-125-NH_2_ (in the region of <500 nm), are eventually carried from their CB to H_2_O/H^+^ where they catalyze the kinetics of hydrogen liberation reactions.

## 7. Modern Developments in Photocatalysts for High-Kinetic Water Splitting under Visible Light

Advances in hydrogen production using visible-light photocatalysts from water have been achieved by combining photocatalyst material with sulfides [88,89], nitrides [82,90], carbides [91,92], and phosphides [93,94]. After combination, a more efficient photocatalyst is achieved that can produce hydrogen at a higher rate. Table 2 documents photocatalysts and sacrificial agents along with a co-catalyst, which form a suitable electronic structure for visible-light activity [46].

In this section, recent developments in visible-light-active photocatalysts via material formulation are reviewed. Material formulation by modern techniques is used to customize the morphology and crystallinity of photocatalysts. The morphology, including structural defects, has a great effect on water splitting under visible light, as discussed in the previous section.

### 7.1. Titanium-Based Photocatalysts

Titanium oxide is regarded as the first semiconductor for photocatalyst-based water splitting [16], but because of a large band gap (3.2 eV) [110], it can only utilize UV radiation [111], which accounts for only a small portion of the solar spectrum. Still, many studies improved upon TiO_2_ photocatalysts. Multiple changes have been made in its structure to enhance the visible-light activity, such as doping with metal ions (Fe^3+^, V^5+^, Co^2+^, Cr^3+^, and Ni^2+^) in the TiO_2_ lattice [112,113]. However, a small increase in efficiency achieved using this technique is insufficient to fulfill the desired efficiency. Kato and Kudo [114] reported the doping of the TiO_2_ photocatalyst at the nanoscale with Sb^5+^ and Cr^3+^, which resulted in the efficient evolution of oxygen from an aqueous solution, using AgNO_3_ as a sacrificial agent under visible light. Cr^3+^ is doped in the TiO_2_ lattice to introduce a new electronic level within the band gap of TiO_2_. This makes the semiconductor sensitive to visible light because of an effective decrease in the band gap, but a problem arises because of unbalanced charge, which is compensated for by co-doping with Sb^5+^ to prevent the formation of Cr^6+^ ions and prevent defects in the lattice caused by oxygen [46]. A thin film of TiO_2_ shows visible-light activity after doping with transition metal ions by modern ion implantation [56,115]. TiO_2_ thin films doped with cations, such as V^+5^ and Cr^+3^, show photo activity under visible light, using CH_3_OH as a sacrificial reagent for water splitting to evaluate hydrogen, having a quantum yield of 1.25 [116]. Ion implantation, a technique of making a wide band gap material into a narrow-band-gap material, is limited by its high cost and low quantum yield [46].

Visible-light activity can also be achieved by doping TiO_2_ with anions [117], such as S [59,118], N [119], and C [56], because p orbitals of these anions get mixed with the 2p orbitals of oxygen present in TiO_2_ lattice which give rise to VB edge upward and band gap decreases [46,92]. Metal titanites, such as SrTiO_3_, LaTi_2_O_7_, and Sm_2_Ti_2_O_7_, are obtained when TiO_2_ reacts with SrO, Ln_2_O_3_ (Ln stands for lanthanides), and BaO, but they have wide band gaps and are unable to absorb visible light. Doping of titanite with Cr^3+^-Ta^5+^ or Cr^3+-^Sb^5+^ on rhodium makes the photocatalyst sensitive to visible light [114]. Using Pt co-catalysts in SrTiO_3_ makes it a visible-light photocatalyst that produces hydrogen from aqueous methanol. La_2_Ti_2_O_7_ doped with Cr^3+^-Fe^3+^ can promote an electron from 3d orbitals of Cr^3+^ or 3d orbitals of Fe^3+^ to the CB, making it photoactive under visible-light irradiation [113]. However, it cannot split pure water because it has a very low redox potential. It can split water if methane is used as electron donor. Another technique used to make Ln_2_Ti_2_O_7_ visible-light photocatalysts up to wavelengths of 650 nm is the partial substitution of oxygen with sulfur anions in the Ln_2_Ti_2_O_7_, such as Sm_2_Ti_2_S_2_O_5_, as shown in Figure 13. The s and 3p valence orbitals of Sm_2_Ti_2_S_2_O_5_ lower the band gap to 2.0 eV from of the 3.5 eV for Sm_2_Ti_2_O_7_. Under visible light, Sm_2_Ti_2_S_2_O_5_ is a stable photocatalyst and can reduce hydrogen ions to hydrogen gas and oxidize the water to oxygen gas using an electron donor sacrificial reagent, such as Na_2_S-Na_2_So_3_ or methanol, or Ag^+^ acceptor ions [46]. Recently, titanium disilicate (TiSi_2_) has emerged as a potential candidate to split the water by acting as prototype photocatalyst under visible-light irradiation [46].

### 7.2. Tantalate- and Niobate-Based Photocatalysts

Structural irregularities present in the tantalates and niobates make them photoactive under UV light because of the high band gap (4.0–4.7 eV). Tantalates and niobates share a corner in the MO_6_ unit (M stands for metals, such as Ta and Nb). They provide a high transfer rate of photogenerated charges and do not allow them to combine again. Hence, under UV irradiation, this can split the pure water into H_2_ and O_2_ [121,122]. Kato and Kudo reported that the metal tantalate MTaO_3_ (M = Li, Na, K) are effective in photocatalytic water splitting [123] and their band gap depends upon cation (using Li = 4.7 eV, Na =4.0 eV, and K = 3.7 eV) [122]. NaTaO_3_ in combination with NiO produces H_2_ and O_2_ from water under UV irradiation to give an efficiency of 20–28% [46]. Teng et al. reported that NaTaO produced via the sol–gel method has high efficiency than any other metal tantalates because of the larger surface area of the product produced and the monoclinic structure of the product achieved using this method instead of orthorhombic with a band gap of (4–4.2 eV) [124].

Among the many approaches to developing a visible-light-active photocatalyst, one is producing oxynitride compounds using tantalates and niobates. In a metal oxide, partial substitution of the oxygen atom with a nitrogen atom causes hybridization between the 2p orbitals of oxygen and 2p orbitals of nitrogen, which cause the VB to shift upward to a higher potential, narrowing the band gap. Following this strategy, Mta_2_N (M = Sr, Ca, Ba), TaON, and Sr_2_Nb_2_O_7-x_N_x_ (X = 1.5–2.8) photocatalysts have been studied under visible light to split water [125,126]. TaON and Mta_2_N (M = Sr, Ca, Ba) show band gaps of 2.5 and 2.5–2.0 eV, respectively, which absorb visible light up to 630 nm. Figure 14 shows the UV/visible spectra of tantalates and niobates. 

TaON photocatalyst produces H_2_ using methanol as a sacrificial reagent and produces O_2_ from an aqueous solution using electron donor AgNO_3_ but Mta_2_N cannot oxidize water because of lower oxidation potential while it can produce H_2_ using an electron donor sacrificial agent (methanol) under visible-light irradiation [46]. Nitrogen-based oxynitrides such as Sr_2_Nb_2_O_7_ oxide can show photoactivity when irradiated with visible light because of the lowering of band gap by hybridization of 2p orbitals of N and with 2p orbitals of O. The higher the content of nitrogen in the oxynitrides higher will be the photoactivity until the original structure of oxynitrides remains unchanged. When the nitrogen content tends to exceed the limit, the original structure of layered Sr_2_Nb_2_O_7_ cannot be maintained and efficiency decreases [46]. 

### 7.3. Other Transition Metal Oxides

Photocatalytic water-splitting reactions can be carried out using specific compounds of vanadium and tungsten such as Ag_3_NO_4_ with a structure similar to that of CaTiO_3_ (perovskite). BiVO_4_ exhibits a structure (sheltie) that is photoactive and can produce oxygen from an aqueous solution of AgNO_3_ under visible-light irradiation [112,123,127]. These oxides have lower band gaps because of the hybridization of the 2p orbitals of oxygen with valence orbitals of BiVO_4_ and Ag_3_VO_4_. Thus, the VB potential rises near the CB to become active under visible light but conduction band electrons of BiVO_4_ and Ag_3_VO_4_ have sufficient potential to reduce 2H^+^ ion to hydrogen gas. WO_3_ is also a useful candidate to evolve oxygen under visible-light irradiation, but only when it is coupled with platinum, which is expensive. Pt-WO_3_ can be used in the presence of NaIO_3_ and Pt/SrTiO_3_, and the water-splitting reaction takes place using monochromatic light with a wavelength of 420.7 nm [128].

### 7.4. Metal Nitrides and Oxynitrides

Transition metal ions with the d^10^ configuration, such as Ga^3+^ [129], react with nitrides and oxynitrides to yield transition metal nitrides [130] and oxynitrides, e.g., ZnO [131] and GaN, reacting to give a solid solution with a composition of (Ga_1-x_Zn_x_)(N_1-x_O_x_) [132], which have band gaps of 2.4–2.8 eV. ZnO and GaN separately have band gap energies greater than 3 eV, but they are modified to respond to visible light as a solid solution. The solid solution of both valence bands contains electrons from the 3d orbitals of Zn and 2p orbitals of N, which results in p-d repulsion [46]. Consequently, the band gap decreases to become active under visible light. However, the solid solution of composition (Ga_1-x_Zn_x_)(N_1-x_O_x_) is modified using superficial deposition of another catalyst (co-catalyst) to give higher visible-light activity [48,133]. Rh and Cr are proven to be efficient co-catalysts for (Ga_1-x_Zn_x_)(N_1-x_O_x_) [134]. This process does not utilize sacrificial reagents to produce H_2_ and O_2_. The quantum efficiency of the solid solution after modification with Rh-Cr is 5.9% under visible light [48]. Powder X-ray diffraction and UV–visible spectra of GaN, GaN:ZnO, GaN:ZnO, GaN:ZnO, and ZnO are shown in Figure 15.

Additionally, the solution of ZnO with ZnGeN_2_ having the composition of (Zn_1-x_Ge)(N_2_O_x_) has been studied. It is also photoactive under visible light with a narrow band gap from parent molecules. Here, also p-d repulsion causes the band gap to narrow. If RuO_2_ is used as a co-catalyst, then the evolution of hydrogen and oxygen takes place stoichiometrically from water [46].

### 7.5. Metal Sulfide Based Catalysts

Metal sulfides are visible-light-active photocatalysts with small band gaps; however; they are unstable in oxidation reactions of water under visible light [134]. Perhaps the reason behind this is that the S^2-^ anion is a more highly reducing agent than water. Thus, S^2-^ is oxidized by the catalyst rather than oxidizing the water, which causes degradation of the photocatalyst [135]. By using sacrificial electron-donating reagents, such as Na_2_S/Na_2_SO_3_ salt, photocatalytic degradation can be prevented [136]. There are many metal sulfides available, among them, wurtzite structured CdS is the most effective photocatalyst [137,138] because of its narrow band gap of 2.4 eV, which is active under visible light and can produce hydrogen and oxygen from water. Possibly, the use of Pt as a co-catalyst can enhance the photoactivity of the catalyst, which results in a higher quantum yield of approximately 25% [139]. Composites of CdS with the semiconductor photocatalysts, such as ZnO [44,140], TiO_2_ [141], and CdO [142], have been studied, where electrons transfer from CdS to other photocatalysts while holes remain in CdS, which facilitates the charge separation and prevents recombination, thus increasing the efficiency of the process [143].

ZnS incorporated into the CdS structure to make a solid solution is also an effective strategy to make visible-light-active photocatalysts for water splitting. Both materials form a continuous sheet in a solid solution with the composition of Cd_1-x_Zn_x_S [144]. Photochemical and photophysical properties of Cd_1-x_Zn_x_S solid solution were studied by Valle et al. [145], using different Zn concentrations (0.2 < X < 0.35). Figure 12 indicates a blue shift of the solid solution. As the concentration of zinc rises from 0.2 to 0.3, the photocatalytic activity of the solid solution (Cd_1-x/_Zn_x_S) increases, as shown in Figure 16.

When the concentration of zinc changes, the band gap is modified because there is a positive change in electronic states between the VB and CB in a solid solution. That is why the evolution rate of H_2_ gas also varies, because structural changes are responsible for the band structure and mobility of photogenerated charge carriers. It is useful to study structural changes in the photophysical properties of Cd_1-x_Zn_x_S solid solution induced using thermal treatment. At high temperatures (113–8733 K), Cd_1-x_Zn_x_S shows an increase in crystalline size. The modification in Cd_1-x_Zn_x_S structure occurs because the degree of Zn substitution increases in the CdS structure. These changes lead to higher crystallinity of the photocatalyst and a high degree of Zn substitution into the CdS structure, resulting in higher catalytic activity of the semiconductors under visible light [146].

Other sulfides, such as ZnS, have also been studied. The band gap of ZnS is not suitable to absorb visible light but doping with various transition metals, such as Ni^2+^, Cu^2+^, Pb^2+^, makes it responsive to visible light. Because of doping, the band gap becomes narrower and photocatalysts become sensitive to visible light for H_2_ production from aqueous solution, using SO_3_^2-^/S^2-^ as a sacrificial electron donating reagent. Other sulfides, such as In^3+^ types [146] and transition metal types, such as Cd^2+^, Cu^2+^, Zn^2+^, and Mn^2+^, have been investigated; however, quantum yield for water splitting using visible light is still limited to 3.7% at 420 nm for the Na_14_In_17_C_43_S_35_ photocatalyst, which is considered most active among the In^3+^ types [146].

## 8. Theoretical Study for Water Splitting Using Nanophotocatalysts

Photocatalytic water splitting utilizing solar energy is a practical and economical solution for hydrogen production as it is a clean and sustainable energy resource [147]. Significant efforts have been made to enhance the solar-to-hydrogen efficiency of nanophotocatalysts. Light harvesting and e/h pair separation are crucial for increasing the process efficiency, which drives the development of novel photocatalytic materials. Recently, theoretical developments in photocatalytic water-splitting material design have gained more attention from scientists designing novel photocatalyst materials. To improve photocatalytic performance, general material design strategies are discussed, such as doping, co-doping, and composite nanometal oxides, reducing material dimension to shorten carrier migration pathways to inhibit charge recombination, and constructing heterojunctions to improve light harvesting and charge separation.

Facing these obstacles, semiconductor band gap engineering is a frequently used approach for developing photocatalysts with improved water-splitting potential. Impurity states are created inside the wide band gap of photocatalysts by doping, without affecting the locations of the VBMs and CBMs, or the chemical reactivities of photocatalysts are enhanced so that the positions of the VBM and CBM may be tweaked further to minimize the band gap [148]. The primary principle for theoretical calculations based on density functional theory (DFT) is a useful tool for materials design [149]. In DFT studies, the mandatory and fundamental step is the optimization of a molecule. Different basis sets are utilized to optimize the molecular structure, such as (B3LYP, LanL2DZ) [150], (CAM-B3LYP, LanL2DZ) [150], etc.

The co-doped TiO_2_ nanomaterial has a large bandgap, but the TiO_2_ nanomaterial has been intensively researched as a photocatalyst for water splitting because of its nontoxicity, cost-effectiveness, and photocorrosion resistance [151]. The fundamental issue in the synthesis of TiO_2_-based photocatalysts is the large bandgap, which only operates when exposed to UV light. It has been demonstrated that doping TiO_2_ with metallic or non-metallic materials effectively reduces the band gap of TiO_2_. Unfortunately, merely doping one element will also diminish carrier mobility and introduce additional recombination hotspots [152]. To adjust the band gap of TiO_2_, an approach of co-doping (two types of atoms) is proposed. Co-doping methods are often classified into three types. The charge compensated by n-p co-doping is the first type [153]. By creating dopant levels near the band boundaries, this method does not alter the pre-existing electronic structures. The second method is known as non-compensated n-p co-doping. The n- and p-type dopants occupy sites near the cation and anions, with uneven charge states. Another method to control the band gap is double-hole-mediated dopant coupling. Many efforts have been made, guided by these techniques, to adjust the bandgap of TiO_2_ by co-doping in the bulk [154,155]. Moreover, the explicit water environment is considered, and the findings show that the estimated band edge sites of TiO_2_ are acceptable for photocatalytic water splitting [156].

The addition of a stable-charge-compensated donor–acceptor pair (Rh-F) to TiO_2_ reduces its bandgap to 2.31 eV [157]. The intermediate band (IB) produced from co-doping is responsible for the band gap reduction. The IB state located below the Fermi level is fully occupied and delocalized, preventing the development of the electron–hole recombination center. In addition, in co-doped TiO_2_, the CBM and VBM locations are suited for photocatalytic TWS. The computed light absorption spectrum shows that the co-doping system can significantly absorb visible light [158].

Another key method to upgrade photocatalytic activity is the production of nanocomposite materials [159]. Semiconductor photocatalyst optical absorption behavior, which is intimately tied to its electronic structure, is a crucial parameter in assessing its photocatalytic capacity for hydrogen generation via water splitting [160]. Spin-polarized DFT+U computations are used to explore the electrical and optical characteristics of a g-C_3_N_4_TiO_2_ heterostructure (Figure 17). All spin-polarized computations are carried out using the Vienna ab-initio simulation package code’s projector-augmented wave pseudopotentials. An equilibrium spacing (1.94 A) and binding energy ($24\;\mathrm{meV}/A_{2}$) indicate that the g-C_3_N_4_TiO_2_ heterostructure is composed of van der Waals forces. Furthermore, the calculated band gap of g-C_3_N_4_TiO_2_ is much lower than TiO_2_ [148]. Consequently, the visible-light sensitivity of the g-C_3_N_4_TiO_2_ heterostructure is significantly enhanced. Furthermore, the expected type-II band alignment would allow electrons to flow from the g-C_3_N_4_ monolayer to the anatase TiO_2_ surface, allowing oxidation as well as redox processes to occur on g-C_3_N_4_ and TiO_2_, respectively. Conclusively, an electric field will be created inside the interface region. The methods described above can aid in the separation of photoexcited carriers and increase hydrogen evolution activity. The methods described above can aid in the separation of photoexcited carriers and increase hydrogen-evolution activity. Furthermore, g-C_3_N_4_TiO_2_ with greater CB minimum energy than TiO_2_ may efficiently create higher-energy electrons to decrease hydrogen ions [161]. Furthermore, the effects of composite distance and the number of g-C_3_N_4_ layers have been thoroughly investigated. The results show that increasing the number of g-C_3_N_4_ layers increases optical absorption over the whole spectrum. When the composite distance is reduced, similar visible-light enhancement is observed [148].

## 9. Conclusions

Various semiconductors based on carbonaceous materials or metal-based oxides, nitrides, sulfides, and carbides have been applied for hydrogen generation, but each has limitations. Various accomplishments have been made to enhance the kinetics of hydrogen evolution by water splitting, including doping, designing hybrid catalysts, usage of sacrificial agents, etc. Using sacrificial reagents increases the research interest to oxidize and reduce the water, resulting in hydrogen production. Considering the process dynamics, the efficiency is dependent on the photocatalyst used. Because of multiple problems associated with the photocatalyst and the unresolved mechanisms of the photocatalyst activity, the quantum yield is only 5.9%. The required improvements include (i) understanding the detailed mechanism of oxidation–reduction of water on the surface of the photocatalyst, (ii) determining when a co-catalyst involves charge transfer mechanisms from the surface of the photocatalyst to the co-catalyst, and (iii) clarifying the surface chemistry of photocatalysts and imperfections on their surfaces. Thus, all photocatalysts require further research. Explaining the effects of impurities on the surface area and other material effects can help reveal more about water splitting. Using synthetic methods, both the electronic structure and reactivity can be modified and investigated. Consequently, the efficiency of the photocatalytic water-splitting process will increase in the future. 

## 10. Key Challenges and Future Perspectives

The photocatalytic splitting of water to generate hydrogen is a promising way to fulfill the growing energy; however, there are several challenges to meeting the desired solar conversion efficiency, for instance: improving the spectral receptive states to harvest more light from the sun, minimizing the charge recombination rate, and overcoming the mismatch of the over-potential value of semiconductor photocatalysts and water. From our perspective, the following approaches can be applied to improve hydrogen production activity:Novel schemes should be developed to maximize the absorption of solar light using low-cost and stable semiconductors with higher panchromatic response. Currently, semiconductors do not offer sufficient light absorption and suffer from e/h pair recombination. Many studies have addressed these challenges, for instance, doping with cations or anions that can alter the band gap of the photocatalysts, thus enabling it to withstand the redox potential of water.Mimicking the natural photocatalytic system by constructing the dual photocatalytic setups is the holy grail of sustainable energy production. In this approach, two semiconductor catalysts are coupled at their electronic level, which offers the suitable potential equivalent to water molecules for enhanced water-splitting kinetics, for example, by fabricating semiconductor composites or designing different heterojunctions. Furthermore, the nanoscale modulation at interfaces of two photocatalytic semiconductors significantly improves the hole–electron segregation and minimizes charge recombination with enhanced charge transference and utilization rate. A thorough understanding of photochemical setups with regard to catalytic interactions at the electronic level and photoactivity is a prerequisite for solar-to-chemical fuel conversion.Furthermore, sufficient knowledge about the mechanism of water splitting is still required, chiefly that related to light absorption and harvesting, charge segregation, charge mobility across the interfaces of semiconductor photocatalysts, and elementary steps during hydrogen generation, for achieving higher solar-to-hydrogen conversion efficiency. As revealed in this review, the modulation of nanostructures improves charge separation and increases the range of the solar spectrum that can be harvested by these semiconductors.

By clarifying the fundamentals of various water-splitting approaches, and the advantages and disadvantages of various modification tactics, highly proficient photocatalytic setups can be designed that harvest a broad solar spectrum at low cost, and thus, solar water splitting can be a sustainable solution for meeting the growing energy demands.

## Figures and Tables

**Figure 1 nanomaterials-13-00546-f001:**
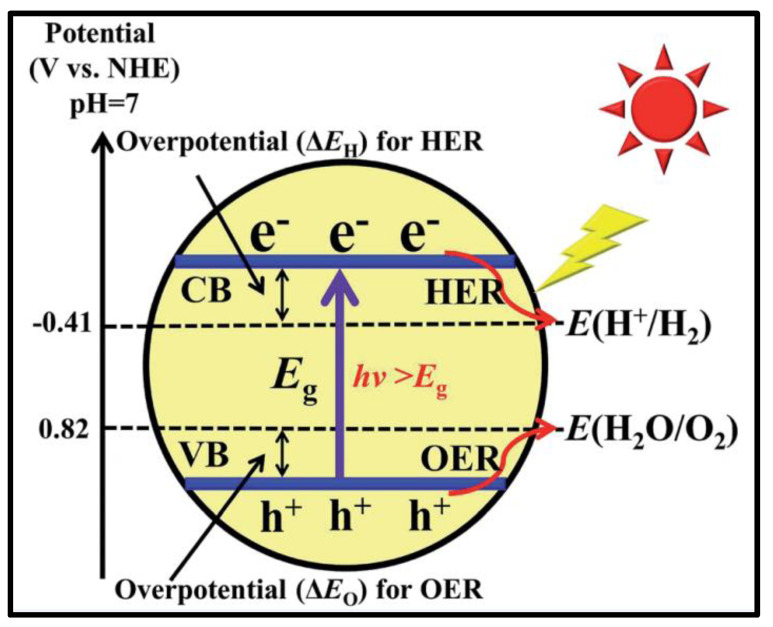
The basic principle of water splitting on the photocatalyst surface. Printed with permission from the Royal Chemical Society [21].

**Figure 2 nanomaterials-13-00546-f002:**
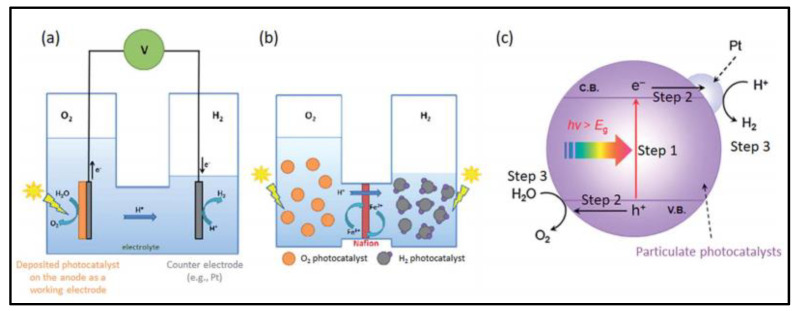
Hydrogen production using (**a**) semiconductor photocatalysts (**b**) photo electrolytic process and (**c**) particular photocatalytic system. Printed with permission from the Royal Chemical Society [42].

**Figure 4 nanomaterials-13-00546-f004:**
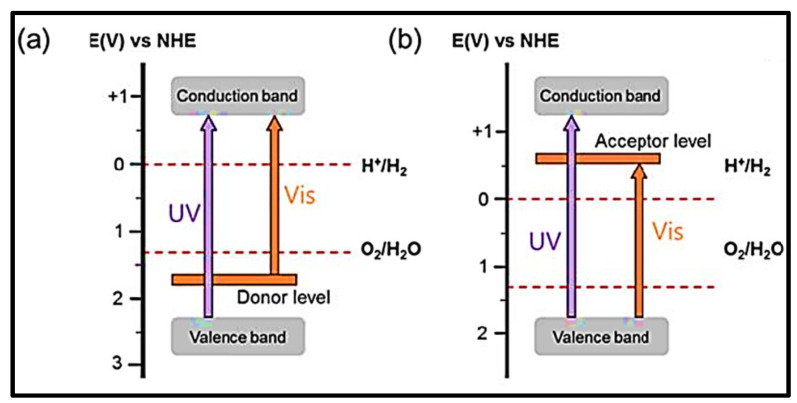
Band gap structure of a cation-doped wide band gap semiconductor. Printed with permission from Wiley Online Library [57].

**Figure 5 nanomaterials-13-00546-f005:**
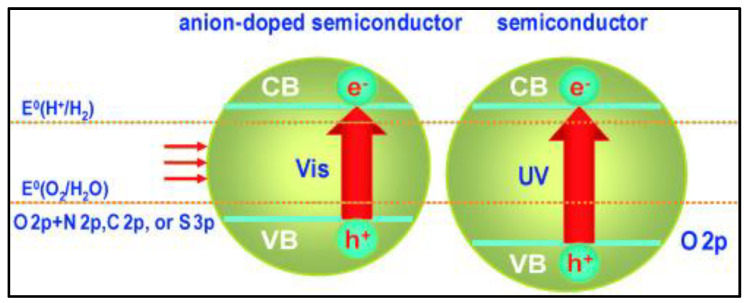
Anion-doped wide band gap semiconductor with visible-light activity. Printed with permission from Wiley Online Library [46].

**Figure 7 nanomaterials-13-00546-f007:**
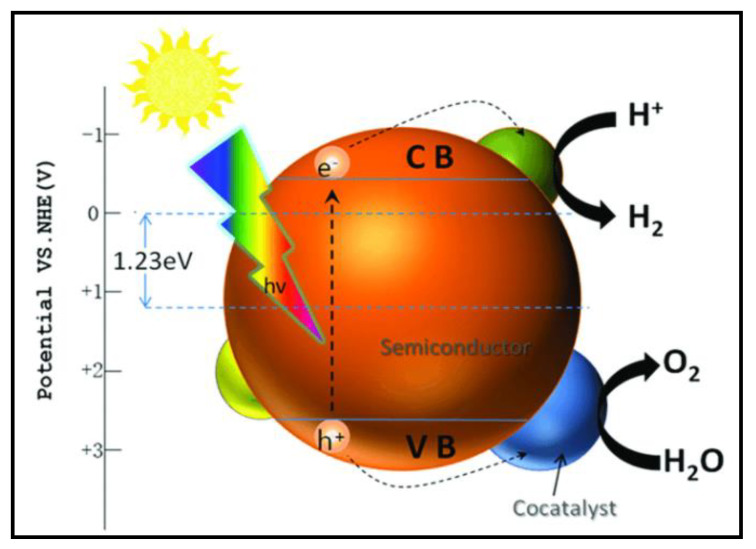
Semiconductor catalyst with co-catalyst. Printed with permission from Frontiers [63].

**Figure 8 nanomaterials-13-00546-f008:**
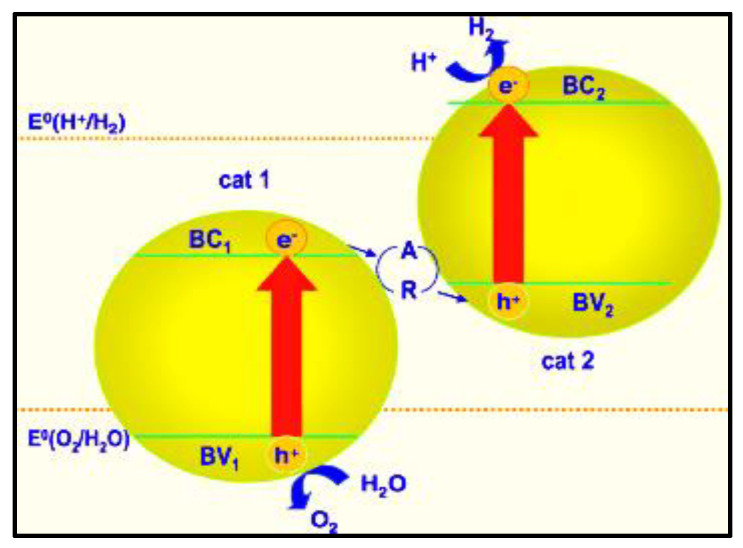
Dual photocatalysts system employing a redox shuttle. Printed with permission from Wiley Online Library [46].

**Figure 9 nanomaterials-13-00546-f009:**
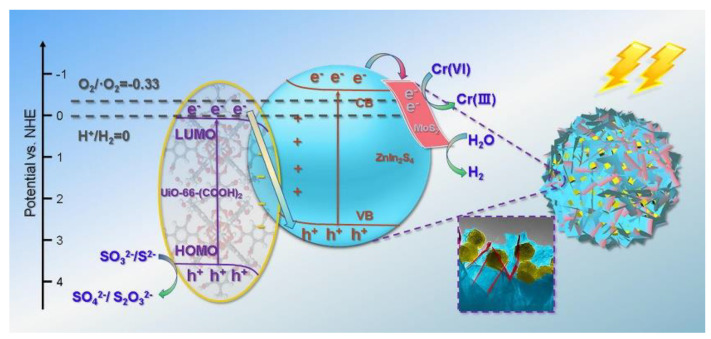
Z-scheme mechanism for photocatalytic water splitting. Printed with permission from Elsevier [80].

**Figure 10 nanomaterials-13-00546-f010:**
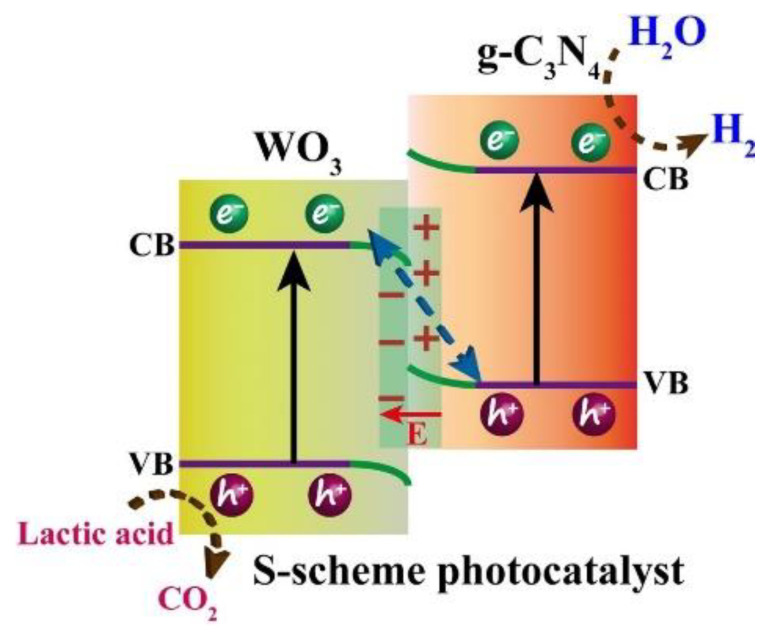
S-scheme pathway for photocatalytic water splitting. Printed with permission from Elsevier [81].

**Figure 11 nanomaterials-13-00546-f011:**
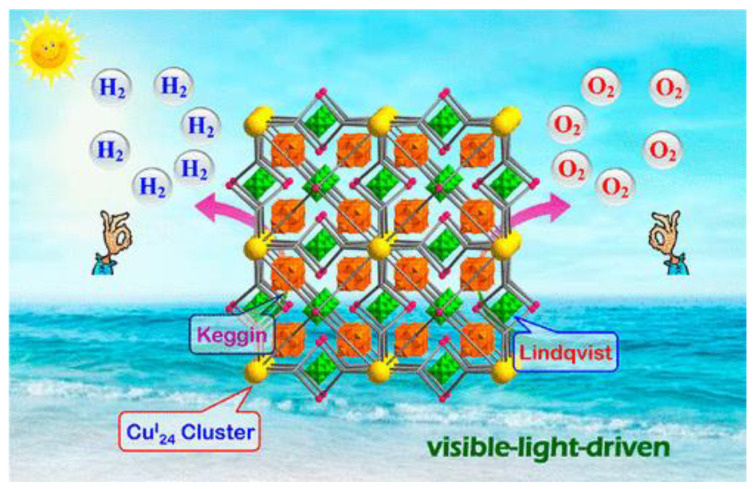
Dual-functionalized mixed Keggin- and Lindqvist-type [Cu^I^_24_(µ_3_-Cl)_8_ µ_4_-Cl]_6_] based POM@MOFs as H_2_ and O_2_ evolution photocatalysts. Printed with permission from the American Chemical Society [86].

**Figure 12 nanomaterials-13-00546-f012:**
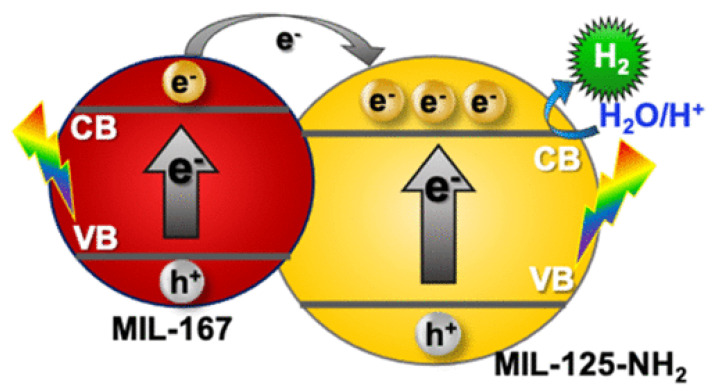
Diagram of an instance of type-II MIL-167 and MIL-125- NH_2_ heterojunctions. Printed with permission from American Chemical Society [87].

**Figure 13 nanomaterials-13-00546-f013:**
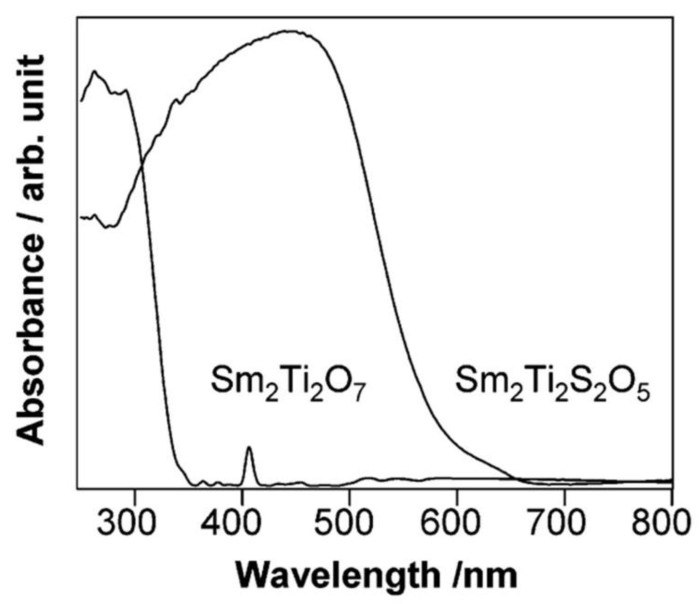
UV/visible diffuse reflectance spectra of Sm_2_Ti_2_O_7_ and Sm_2_Ti_2_S_2_O_5_. Printed with permission from American Chemical Society [120].

**Figure 14 nanomaterials-13-00546-f014:**
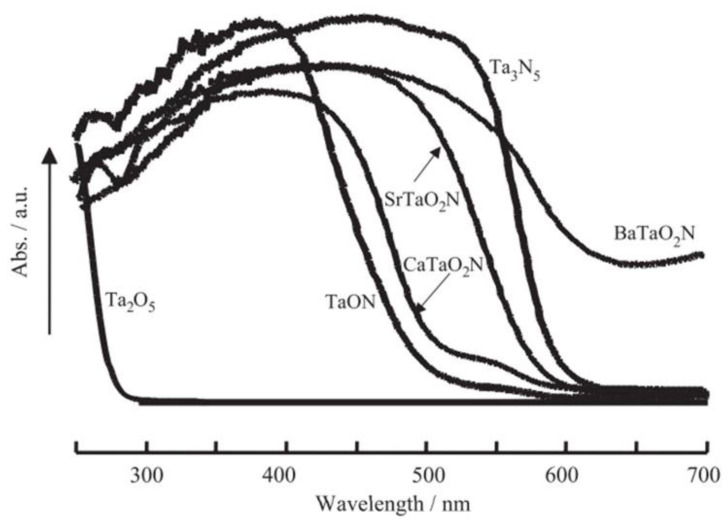
UV–visible diffuse reflectance spectra of Ta_2_O_5_, TaON, Ta_2_N_3_, and MtaO_2_N (M: Ca, Sr, Ba). Printed with permission from Elsevier [125].

**Figure 15 nanomaterials-13-00546-f015:**
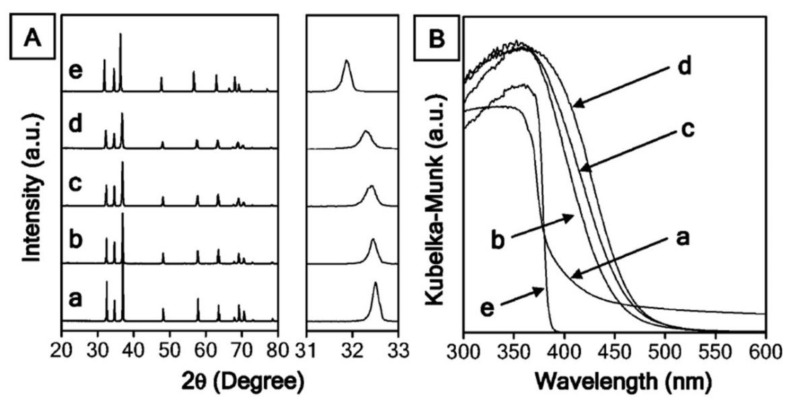
(**A**) Powder X-ray diffraction patterns and (**B**) UV/visible diffuse reflectance spectra of (a) GaN, (b) GaN:ZnO, (c) GaN:ZnO, (d) GaN:ZnO, and (e) ZnO. Printed with permission from American Chemical Society [132].

**Figure 16 nanomaterials-13-00546-f016:**
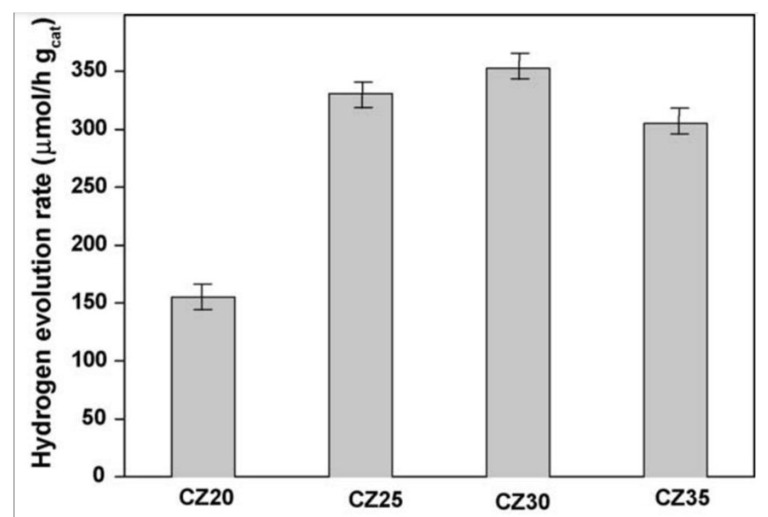
Hydrogen production rate from an aqueous solution of Na_2_S and Na_2_SO_3_ in visible light over solid solutions of Cd_1-x_Zn_x_S with different zinc concentrations (0.2, 0.25, 0.30, and 0.35) in 150 mL of reactant solution with a 0.1 M concentration of Na_2_S and 0.04 M concentration of Na_2_SO_3_. Printed with permission from Elsevier [145].

**Figure 17 nanomaterials-13-00546-f017:**
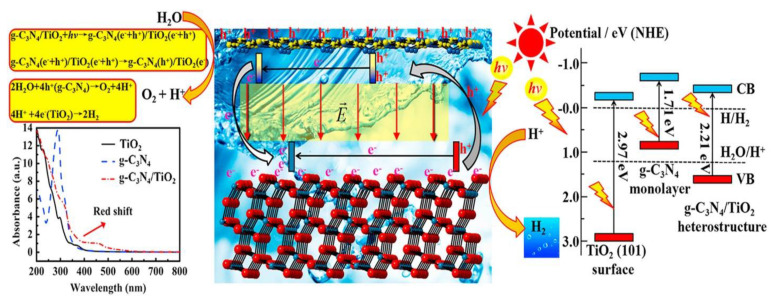
Schematic diagram of g-C_3_N_4_TiO_2_ band gap reduction and working efficiency. Printed with permission from Elsevier [148].

**Table 1 nanomaterials-13-00546-t001:** Distribution of energy in the solar spectrum.

Scheme 315	Near UV	Blue	Green/Yellow	Red	Near IR	IR
**Wavelength (nm)**	315–400	400–510	510–610	610–700	700–920	920–1400
**Energy (eV)**	3.93–3.09	3.09–2.42	2.42–2.03	2.03–1.77	1.77–1.34	1.34–0.88
**Contribution to total spectrum (%)**	2.9	14.6	16.0	13.8	23.5	29.4

**Table 2 nanomaterials-13-00546-t002:** Summary recently developed for visible-light-active photocatalyst.

Photocatalyst	Sacrifice Agent	Synthesis Method	Morphology	Light Source	Band Gap	H_2_ Evolution	Ref.
**CN@ZnIn_2_S_4_**	Na_2_S/Na_2_S_3_	---	Ultra-thin nanosheets	Xe lamp (300 W)	---	3.17 mmol g^−1^h^−1^	[95]
**PbTiO_3_-TiO_2_**	---	Hydrolysis–hydrothermal method	Octahedral	UV light	2.65	630.51 μmol g^-1^h^−1^	[96]
**Fe SrTiO_3_**	---	One-step hydrothermal route	Cubic-like particles	UV–VIS	1.61	1376 μmol g^-1^h^-1^	[97]
**CaTiO_3_**	---	Impregnation method	Nanoparticles	Xe lamp (300 W)	3.4	0.39 μmol min^-1^	[98]
**Co/NGC/ZnIn_2_S_4_**	TEOA	In situ solution growth method	Nanosheets	Xe lamp (300 W)	2.1	11.27 mmol g^−1^h^−1^	[99]
**TiO_2_**	Na_2_SO_3_	Hydrothermal	Nanorod	Visible light	2.4	100 μmol	[100]
**CoOTiO_2_**	Na_2_SO_3_	Photochemical deposition and thermal decomposition	Needles	Visible light	2.6	540,000 μmol h^−1^cm^−2^	[101]
**CaTiO_3_/Cu/TiO_2_**	Na_2_SO_4_	Hydrothermal reaction	Groove structures	Xe lamp (300 W)	3.37	23.55 mmol g^-1^h^−1^	[102]
**La,Al-Codoped SrTiO_3_**	------	Flux treatment method	Core–shell structure	Xe lamp (300 W)	2.25	1790 μmol g^-1^h^−1^	[103]
**H-doped TiO_2_**	NaOH	Hydrothermal	Nano-bullets	Visible light	3.05	81.3 μmol h^−1^	[104]
**TiO_2_BiVO_4_**	NaOH	Wet chemical	----	Visible light	2.64	6 μmol h^−1^	[105]
**Pt-doped TiO_2_**	CH_3_OH	Direct hydrolysis	Rods	Visible-light-simulated solar light	2.74	932/1954 μmol g^-1^h^−1^	[106]
**Ag-TiO_2_**	NaHCO_3_	Sol–gel and metal–organic decomposition	---	Visible light	---	1070 μmol min^-1^	[107]
**Ru-doped LaFeO_3_**	Na_2_SO_4_	Solid-state reaction	small spherical particles	UV-Vis	3.4	1133 μmol g^-1^h^−1^	[108]
**ZnO/CdS**	S^2−^ and SO_3_^2−^	Hydrothermal method	heteroepitaxial	300 W Xe lamp	2.5	669.6 μmol/h	[109]

## Data Availability

Data will be available on request.

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
