# Peer review of "Semiconductor Nanomaterial Photocatalysts for Water-Splitting Hydrogen Production: The Holy Grail of Converting Solar Energy to Fuel"

_nanomaterials, 2023, doi:10.3390/nano13030546_

Round 1
Reviewer 1 Report
The manuscript entitled “Semiconductor nanomaterials photocatalysts for water splitting hydrogen production: A holy grail for solar energy to fuel conversion" is an interesting approach to describe the present state of the knowledge about the application of various photocatalytic materials for water splitting hydrogen production. This manuscript can be considered for the publication in Nanomaterials after the major revision including the following remarks:
1. Authors should more express the novelty and need for this review in the context of other reviews in this field, recently published, e.g., Int. J. Hydr. Energy, 44, 2019, 540; Int. Hydr. Energy, 48, 2023, 523; Int. J. Energy Res., 46, 2022, 5467; ECS Adv., 1, 2022, 030501, ACS Energy Lett., 4, 2019, 1687.
2. Conclusion section should be extended to include the perspectives and point out the main challenges in this field.
3. The quality of many figures should be improved, e.g., 1, 4, 5, 6, 8, 12.
4. The editing issues and the English language of the manuscript should be extensively improved.
Author Response
We have made revisions to the manuscript as per your suggestions. Kindly see the attached detailed response letter.

Reviewer 2 Report
Journal: nanomaterials
Manuscript number: nanomaterials -2137549
Article type: Review
Title: Semiconductor nanomaterial photocatalysts for water splitting hydrogen production: A holy grail for solar energy to fuel conversion
Nanomaterials attracted the scientific research community for photocatalytic hydrogen production due to its extraordinary properties such as high specific surface area, attractive morphology and high absorption ability. The present review highlights various modification strategies accessible to boost up the hydrogen production by photocatalytic splitting water and describes various endeavors which have been made by the scientists to upsurge the kinetics of hydrogen production. Major revision is suggested to further improve its quality. Specific suggestions are provided below.
1. There are some typo and grammar errors. This will lead to a difficult understanding of readers on what authors planned to express.
2. In the section 2, Different methods to produce hydrogen. That doesn't quite fit the title.
3. I didn’t see some new merits and challenges in section 4.
4. In the section 5” Effective way to engineering efficient photocatalysts”, some new ideas can be borrowed from Applied Catalysis B: Environmental 2020. 273: 119051 and Chemical Engineering Journal 2021. 404: 126540.
5. The authors can add a section about theoretical calculations.
6. The conclusion is too simple.
Author Response

(The authors gave the same response as above.)

Reviewer 3 Report
Dear Authors! Thank you for your manuscript, submitted to Nanomaterials. I have read your Review Article, devoted to very actual topic, connected with creation of photocatalysts for water splitting hydrogen production.
I would like to say, that according to Scopus, at the present moment 181 Reviews were published on the above-mentioned problem only for 2020-2023 period. So, it is very difficult nowadays to identify research gap in the considered topic. On this reason, I have the serious comments to your manuscript:
1. What did the new strategies for the photocatalysts development appeared, in comparison with Review from 2009 (Ref. [17])?
Your manuscript does not consider the modern and newest achievements in the fields of photocatalysts: surface coatings, nanohybrids, metal-organic framework, 2D-materials, semiconducting polymers and so on. Your manuscript does not notice about the Z-scheme and S-scheme of heterojunctions. In fact, your manuscript is devoted to the photocatalyst TiO2 only. Therefore, your manuscript looks outdated.
2. The works in manuscript are cited randomly.
There are many non-relevant reference, for example:
Introduction. [1-5] - Refs [1-4] are not devoted to environmental pollution; Refs [6], [7] are not about the greenhouse gases; Ref [9] - about the photocatalysts, not about the energy requirements; Refs [10], [11], [16] - about catalysts, not about the energy demand; Ref [13] is strange in context; "recent literature 12, 17" - Ref [12] - 2014; Ref [17] - 2009, are they recent? What is the reference by Honda and Fujishima?
3. Paragraph 5.1 does not contain references.
4. 5.5 Nanostructure. Refs [69], [70] are not about sol-gel; [72] is not about hydrothermal method; [78] is not about zeolits; [81] is not about photocatalysis. Refs [62-82] are very old (1984-2013 period).
5. Table 2 contains completely very old references: [85] of 1995; [99-112] -of 2000-2005
6. It is necessary to write the Copyright permissions correctly (for ex., from the permission of Elsevier ..).
7. What is "Fig.3 [36]"? (last paragraph at Page 4).
What is "Fig. 6" ? (last paragraph at Page 8).
8. Are the figures 2c and 3 the same?
So, I consider that reviewed manuscript needs in significant rewritting.
Author Response

(The authors gave the same response as above.)

Round 2
Reviewer 1 Report
After corrections, the manuscript in the present form can be accepted for the publication.
Author Response
Kindly see the attached detailed response

Reviewer 2 Report
Accept in present form.
Author Response
Kindly see the attached detailed response .

Reviewer 3 Report
Dear Authors! Thank you for your great attention to my comments. I see, that your work was expanded strongly with the material adding, besides the references were renewed. English was corrected in initial parts, but new parts are needed in Eng. proof-reading too.
Moreover, manuscript in the present form is not ready for publication, because, the innovation of the review should be shown strongly - how does the presented manuscript differ from the same-topic reviews, for example, Refs [20] -recent review by Authors and [46]. The structure of manuscript should be more visible and clear. Abstract should contain the main ideas of manuscript.
Besides, it is necessary to restore the order in the references and figures.
For example, the Refs. [46] and [80] are the same.
Fig. 16. Original experimental data is in https://doi.org/10.1016/j.cattod.2008.09.024
Fig. 15. Ref it is not [129]
Fig. 14. Fig was adapted from [129]
Fig. 13. Original experimental data is in https://doi.org/10.1021/ja0269643
Fig. 12. Publisher is not Elsevier. It is ACS.
My recommendation is to choose figures for Review more carefully. There should be no copies of figures from the another reviews. It is better to use original data with your own analysis and drawings.
It is necessary to control and mark in different ways figures of Open Access and by Copyright.
Author Response

(The authors gave the same response as above.)

Round 3
Reviewer 3 Report
Dear Authors! Thanks for your attention to my comments.